# Environmental Exposures during Puberty: Window of Breast Cancer Risk and Epigenetic Damage

**DOI:** 10.3390/ijerph17020493

**Published:** 2020-01-13

**Authors:** Rama Natarajan, Dana Aljaber, Dawn Au, Christine Thai, Angelica Sanchez, Alan Nunez, Cristal Resto, Tanya Chavez, Marta M. Jankowska, Tarik Benmarhnia, Jiue-An Yang, Veronica Jones, Jerneja Tomsic, Jeannine S. McCune, Christopher Sistrunk, Stacey Doan, Mayra Serrano, Robert D. Cardiff, Eric C. Dietze, Victoria L. Seewaldt

**Affiliations:** 1City of Hope Comprehensive Cancer Center, Duarte, CA 91010, USA; rnatarajan@coh.org (R.N.); daljaberz1996@yahoo.com (D.A.); dau@coh.org (D.A.); chthai@coh.org (C.T.); angsanchez@coh.org (A.S.); anunez@coh.org (A.N.); cresto@coh.org (C.R.); tachavez@coh.org (T.C.); vjones@coh.org (V.J.); jtomsic@coh.org (J.T.); jmccune@coh.org (J.S.M.); csistrunk@coh.org (C.S.); staceyndoan@gmail.com (S.D.); maserrano@coh.org (M.S.); rdcardiff@ucdavis.edu (R.D.C.); edietze@coh.org (E.C.D.); 2City of Hope Diabetes Metabolism Research Institute, Duarte, CA 91010, USA; 3Qualcomm Institute/Calit2, University of California at San Diego, San Diego, CA 92093, USA; majankowska@eng.ucsd.edu (M.M.J.); jayyang@eng.ucsd.edu (J.-A.Y.); 4Family Medicine and Public Health San Diego, University of California at San Diego, San Diego, CA 92093, USA; tbenmarhnia@ucsd.edu; 5Department of Psychology, Claremont McKenna College, Claremont, CA 91711, USA; 6Center for Comparative Medicine, University of California at Davis, Davis, CA 95616, USA

**Keywords:** environment, breast cancer risk, empowerment

## Abstract

During puberty, a woman’s breasts are vulnerable to environmental damage (“window of vulnerability”). Early exposure to environmental carcinogens, endocrine disruptors, and unhealthy foods (refined sugar, processed fats, food additives) are hypothesized to promote molecular damage that increases breast cancer risk. However, prospective human studies are difficult to perform and effective interventions to prevent these early exposures are lacking. It is difficult to prevent environmental exposures during puberty. Specifically, young women are repeatedly exposed to media messaging that promotes unhealthy foods. Young women living in disadvantaged neighborhoods experience additional challenges including a lack of access to healthy food and exposure to contaminated air, water, and soil. The purpose of this review is to gather information on potential exposures during puberty. In future directions, this information will be used to help elementary/middle-school girls to identify and quantitate environmental exposures and develop cost-effective strategies to reduce exposures.

## 1. Introduction

During puberty, the human breast bud undergoes major structural and cellular changes. Many human and animal studies provide evidence that during puberty, the breast is highly susceptible to damage from exposure to environmental toxins (window of susceptibility). Recent studies provide evidence that environmental exposures during puberty may increase the risk of breast cancer in adulthood. Currently, young women are exposed to unhealthy food and environmental toxins. In California, many young women live in rural, suburban, and urban neighborhoods that lead to additional exposure risks.

We are all impacted by environmental exposures. The majority of rural residents lack access to the food they grow and live in food deserts (no grocery stores within 10 miles rural/1 mile urban that provide fresh vegetables, grains, and non-processed meats). With increasing drought, water tables are dropping and there is an increasing concentration of heavy metals in both water and food. Similar to rural women, urban women have a high likelihood of living in neighborhoods that lack access to healthy food. Urban neighborhoods frequently have old plumbing, increasing the risk of heavy metal contamination. The majority of our cities are contaminated with industrial waste.

There are many studies underway investigating the link between poor diet, environmental exposures, and breast cancer. Much is likely to be learned in the future. However, in the present, a generation of young women are being exposed to environmental toxins. Here, we have gathered a panel of experts to discuss some of the potential linkages between environmental exposures, diet, and breast cancer risk. It is the authors’ hope that this discussion can promote strategic interventions to reduce breast cancer risk in young women and help foster a generation of young women who are empowered to act as advocates for their health.

## 2. Puberty and Susceptibility

The mature human breast is composed of milk-producing lobules, connected to the nipple by a system of branching “tree-like” ducts surrounded by adipose and connective tissue. Breast cancer is thought to develop within a terminal ductal lobular unit (TDLU), which includes the lobule and its most proximal ducts [1]. The breast is unique in that full maturation of the breast does not occur until later in life, after puberty. During puberty the female breast undergoes rapid changes: proliferating terminal end buds and elongation of the ducts promote the formation of primitive lobular structures and growth of the ductal tree [2,3] (Figure 1A). These rapid structural and cellular changes are thought to create a “window of susceptibility” (Figure 1A) [4].

Animal and epidemiological studies provide evidence that there are windows of susceptibility: implantation, fetal growth, puberty, pregnancy, and aging, during which, the breast is thought to be particularly vulnerable to environmental exposures. Environmental exposures during windows of susceptibility are hypothesized to increase subsequent breast cancer risk. There is increasing evidence that a woman’s lifetime breast cancer risk is increased by exposures before and during puberty. Epidemiologic studies provide evidence that medications [5] and disease [6,7] before and during puberty, increase subsequent breast cancer risk; studies on breast cancer risk and environmental exposures before and during puberty are both difficult and important. Increasingly, prospective cohorts are being designed to investigate events that occur during this key window of susceptibility [4].

### 2.1. Puberty

Epidemiologic and laboratory studies provide evidence that environmental exposures during puberty are associated with an increased adult risk of breast cancer. In experimental studies on rats, exposure to the carcinogen dimethylbenz [a] anthracene (DMBA) during puberty resulted in the highest number of tumors (vs. other times of life) [4]. In transgenic mice, exposure to high estrogens during puberty resulted in mammary hyperplasia and breast cancers [8,9]. In humans, exposures during puberty to (1) high doses of radiation (Hodgkin’s Disease, atomic bomb) [10,11] and (2) dichlorodiphenyltrichloroethane (DDT) [12,13] both increase breast cancer risk.

### 2.2. High-Risk Field

In 1996, the late Helene Smith proposed a model of mammary carcinogenesis, where breast cancer developed in a “high-risk epithelial field” containing genetic (now also epigenetic) damage that promoted the survival of a second genetic “hit” (Figure 1B) [1]. This model was based on the observation that while (1) invasive breast cancers and their normal appearing adjacent TDLUs both contained characteristic genetic losses, (2) the distal TDLUs lacked these genetic losses (Figure 1B) [1]. Based on these observations, it is hypothesized that (1) specific genetic or epigenetic alterations may occur in a breast progenitor cell prior to or during puberty, and (2) the clonal expansion of a damaged progenitor during puberty would result in a localized predisposed TDLU or “high-risk” field from which breast cancer subsequently would arise (Figure 1B). The high-risk epithelial field hypothesis represents an important breakthrough in our understanding of the potential impact of environmental exposures during windows of susceptibility for breast cancer.

### 2.3. Impact of Delayed Pregnancies

Breast cancer is the most common malignancy diagnosed during pregnancy [14]. Women are electing to have children at a later age; due to this delay in childbearing, cancer diagnosed during pregnancy is likely to rise [14].

**Figure 1 ijerph-17-00493-f001:**
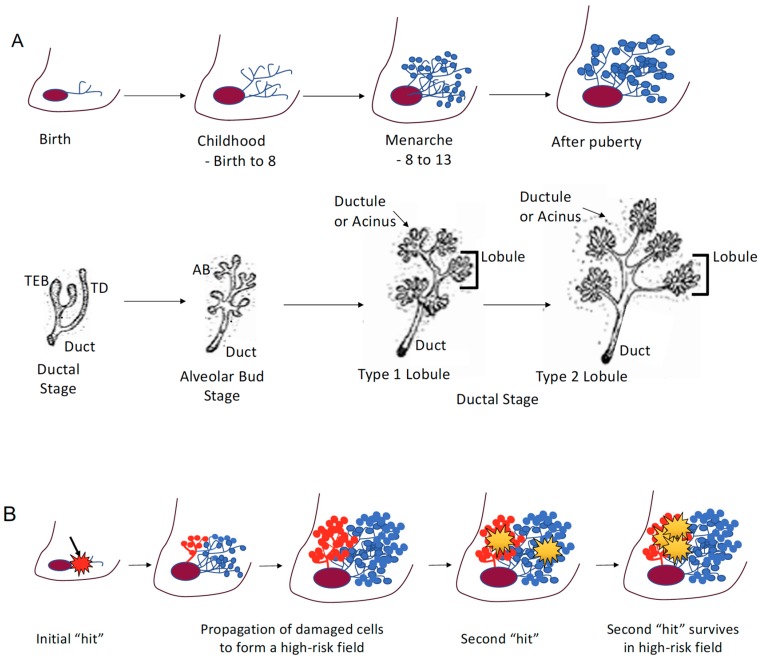
(**A**) Stages of breast development. The breast is unique in that full maturation of the breast does not occur until later in life, after puberty. During puberty, the female breast undergoes rapid changes: proliferating terminal end buds and elongation of the ducts promote the formation of primitive lobular structures and growth of the ductal tree. Breast development illustrations courtesy of the University of California Davis School of Medicine, Robert Cardiff MD, PhD, and Hannah Jensen MD (open source information) [15]. (**B**) The high-risk field hypothesis is that specific genetic or epigenetic alterations may occur in a breast progenitor cell prior to, or during puberty, and the clonal expansion of a damaged progenitor during puberty would result in a localized predisposed terminal ductal lobular unit (TDLU) or “high-risk” field, from which breast cancer subsequently would arise. A second genetic or epigenetic “hit” in a high-risk field is hypothesized to survive and propagate with a higher frequency than a similar “hit” in a normal breast lobule.

## 3. Epigenetics

Epigenetics refers to heritable changes in gene expression and phenotypes without changes in the underlying DNA sequence. This is achieved by special marks added to the DNA or to proteins surrounding the DNA. Epigenetic changes can occur in response to environmental cues and alter chromatin structure and expression of genes associated with pathophysiological conditions, mediate transgenerational effects, and impact risk for cancer and diabetes [16,17,18]. Epigenetic changes can regulate gene programs required for the determination of cell fate during development and maintenance of cell identity and have heritable transgenerational effects [16,19].

### 3.1. Epigenetic Changes Associated with Breast Cancer

The discovery that germline mutations of *BRCA1* and *BRCA2* significantly increase breast cancer was a landmark finding. Despite the importance of this discovery, the majority of breast cancers occur in women who lack germline mutations; increasingly, epigenetic alterations are being implicated in breast cancer initiation and progression. Study of the potential link between breast cancer and epigenetics is just beginning [20].

The epigenetic inactivation of BRCA1/2: Homologous recombination deficiency can result from epigenetic processes, leading to the silencing of homologous recombination genes, including BRCA1/2 [21]. Early studies have shown significant efficacy for PARP inhibitors in patients with BRCA1/2 mutations [20]. It is not clear whether the biological effects of harboring somatic BRCA1/2 mutations are identical to their germline counterparts (BRCAness phenomenon) [21].

Epigenetic mechanisms: Nuclear DNA is packed into chromatin, a histone-protein complex. The basic subunit of chromatin is the nucleosome which is composed of octameric histone protein complexes formed by dimers of core histones H2A, H2B, H3, and H4 and wrapped by 147 bp of DNA. Dynamic changes in chromatin structure determine whether gene expression is “turned on” (activation) or “turned off” (repression) in response to intracellular and extracellular cues. Changes in chromatin structure are regulated by epigenetic modifications such as DNA methylation (DNAme) and histone post-translational modifications (PTMs) [22,23]. In association with non-coding RNAs like microRNAs (miRNAs) and long non-coding RNAs (lncRNAs), these modifications regulate epigenetic mechanisms of gene transcription (Figure 2).

### 3.2. Writers/Erasers

Epigenetic modifications are mediated by “writers” and removed by “erasers”. Epigenetic modifications are interpreted by “readers” that specifically interact with DNAme and modified or unmodified histones [22,23]. DNAme is the best studied epigenetic mark. DNAme at promoters is usually associated with the inhibition of gene expression. DNA methyltransferase 3A (DNMT3A) and DNMT3B mediate *de novo* DNAme patterns during development and are maintained by DNMT1 later in life. DNA demethylases such as ten-eleven translocation enzymes (TET) catalyze the oxidation of 5-methylcytosine and dynamically regulate DNAme to generate 5’-hydroxymethylated cytosine [23].

### 3.3. Histone Modification

Histone post-translational modifications either activate or inhibit gene expression. Histone acetylation and methylation are mediated by histone acetyl transferases (HATs) and histone methyltransferases (HMTs), respectively. Their actions are opposed by histone deacetylates (HDACs) and demethylases (HDMs), respectively. HMTs and HDMs are specific in terms of the particular lysine residue and the extent of methylation. HATs and HDACs lack this specificity [22]. These enzymes also modify lysines on non-histone proteins, including transcription factors [24]. Mechanistically, promoter DNAme or repressive histone post translational modifications repress gene expression by recruiting corepressors and condensing chromatin to form heterochromatin. In contrast, permissive histone PTMs drive gene expression by recruiting coactivators and chromatin remodeling proteins that increase chromatin access and open chromatin formation [22,23].

### 3.4. LncRNAs

Epigenetic mechanisms are fine-tuned by lncRNAs: >200 nucleotide-long transcripts without protein-coding potential. lncRNAs are expressed as divergent, anti-sense, and intergenic transcripts that can host genes of non-coding miRNAs and enhancer RNAs [25]. LncRNAs can exert cellular effects via diverse, cell- and location-specific mechanisms, many of which are still not well understood. Nuclear lncRNAs regulate epigenetic mechanisms to fine-tune the expression of key target genes in *cis* and *trans* via interaction with key RNA binding proteins, epigenetic factors, enhancers, and chromatin. lncRNA-mediated epigenetic mechanisms regulate diverse biological processes, including X chromosome inactivation, genomic imprinting, stem cell differentiation, and inflammation; dysregulation has been reported in association with diabetes and cancer [25].

### 3.5. Imprinting

Genomic imprinting is an inherited form of parent-of-origin-dependent epigenetic gene regulation that renders autosomal genes functionally haploid in a species, developmental stage, and tissue dependent manner [26,27]. There is evidence that epigenetic modifications in the genome link environmental exposures to adult disease susceptibility [28,29,30,31,32]. Imprinting can be dysregulated in germ cells, potentially affecting offspring via parental exposures [29,33]. Because imprinted genes are frequently clustered and coordinately regulated by differentially methylated regions (DMRs), changes in a single DMR can disrupt the expression of many imprinted genes [34,35]. Diet-derived methyl donors and co-factors are necessary for the synthesis of *S*-adenosylmethionine (SAM) (methyl group donor for DNA methylation). Environmental factors that alter early nutrition and/or SAM production alter CpG methylation at important DMRs and impact future disease risk [28,36]. Disease susceptibility due to epigenetic deregulation has specific windows of vulnerability, including embryogenesis, puberty, and pregnancy [28,36].

### 3.6. Gene-Epigenenome Interactions

The epigenome represents the genome-wide interactions among all three epigenetic layers. Epigenomics refers to genome-wide profiles and consequences of these epigenetic marks. Through landmark studies such as the encyclopedia of DNA elements (ENCODE) and the National Institute of Health (NIH) Roadmap Epigenomics Project, high-throughput next-generation DNA sequencing (seq) technologies were developed to investigate the epigenome. These studies have allowed researchers to build high-resolution maps of the chromatin structure and genome-wide distribution of epigenetic modifications at regulatory elements [37]. These technologies as well as the availability of archived DNA samples from various clinical trials have facilitated several epigenome-wide association studies (EWAS) that, in conjunction with genome-wide association studies (GWAS) studies, can shed new light on the causes of common diseases such as breast cancer [38].

**Figure 2 ijerph-17-00493-f002:**
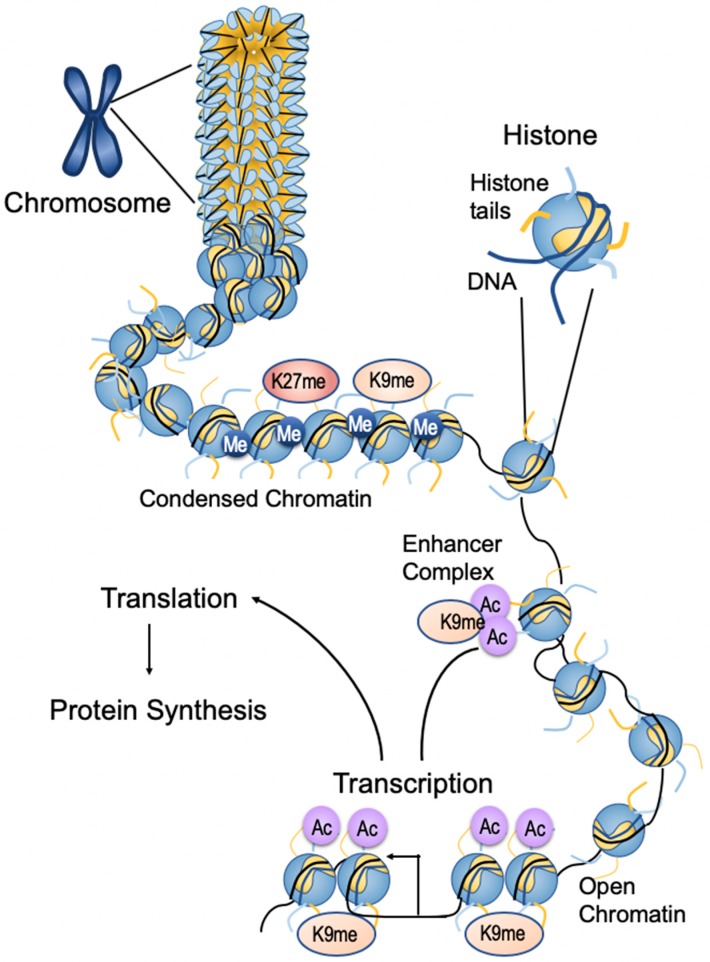
Epigenetic modifications of chromatin structure. Chromosomes are composed of DNA–protein complexes called chromatin. The basic subunit of chromatin is the nucleosome, which comprises an octamer of two copies of each of the core histone proteins—H2A, H2B, H3, and H4—wrapped by 147 bp of chromosomal DNA. Condensed chromatin is generally characterized by promoter DNA methylation and histone methylation, depending on the site of modification on the histone (for example, lysine 9 methylation (K9Me) and lysine 27 methylation (K27Me) on histone H3), resulting in repressed gene expression. Relaxed chromatin is marked by histone lysine acetylation and methylation of lysine 4 on histone 3 (K4Me) in the promoter regions, that promote gene expression. Histone lysine acetylation at the enhancer region also enhances gene expression by interacting with the promoter region through bromodomain-containing proteins (BRD) and other proteins. RNA methylation is a form of epigenetic regulation related to the translation and degradation of RNAs. Figure adapted from Kato et al. [38].

## 4. Exposures

Young women face exposures to carcinogens and potential carcinogens by drinking water, breathing air, applying cosmetics, and eating food (Figure 3). Increasing evidence suggests that early life stress may also impact a woman’s future health. While some compounds such as heavy metals have known toxic effects, much is not known. One key gap in our understanding of how environmental exposures before and during puberty may impact breast cancer risk is that potential environmental carcinogens are typically studied as single agents and not in mixtures. Here, we review some of the major environmental carcinogens and exposures young women may have before and after puberty. Economically disadvantaged neighborhoods disproportionately bear the impact of environmental exposures. This unequal burden of exposures in young women is thought to increase lifelong risk for cardiovascular disease, obesity, and cancers, including breast cancer.

### 4.1. Water

Water is the source of many environmental exposures [39]. Environmental exposures range from biological exposures (bacteria, fungi) to radiation (uranium) to heavy metals (lead, cadmium, arsenic, mercury) [40]. Heavy metals are some of the most dangerous and best studied environmental contaminants. The lack of safe water is increasingly becoming a burden faced by disadvantaged neighborhoods; United Nations Secretary-General Ban Ki-moon stated that “hundreds of millions worldwide still live with an absence of clean water, perpetuating poverty” [41]. From Flint, Michigan to Gaza, disadvantaged neighborhoods and peoples lack access to safe water and face an increasing burden of environmental exposures [42,43].

Heavy metal contamination come from a variety of sources including both point sources (mining) and nonpoint sources (lead pipes). The main causes of water contamination come from metal refineries, cigarette smoke, improper disposal of waste (electronics/batteries), industrial discharge, and leeching from processing sites [44,45].

Heavy metals are strongly linked to carcinogenesis and have been associated with epigenetic modifications in mice and humans [46,47]. Lead, along with other metals (such as chromium), has been seen to induce oxidative stress in cells by altering the cells epigenome [46,48]. Cadmium exposure increased DNA methylation of imprint control regions by 10% in mother/child pairs [49]. Another study connected placental cadmium to DNA methylation of loci regulating metabolism and inflammation [50].

### 4.2. Air Pollution

Air pollution is a known carcinogen, resulting from a mixture of natural substances and man-made emissions [51]. There are two categories: outdoor and indoor air pollution. The focus on outdoor air pollution is related to potential associations between zip-code-related air pollution levels and breast cancer development due to early age exposure.

Since the Clean Air Act of 1970, air quality significantly improved beginning in 2000 with a 16% decrease in unhealthy ozone levels [51]. Although in recent years, there has been an uptick in the number of unhealthy ozone days [51]. Currently, 124 million people living in 201 U.S. nonattainment counties (not meeting EPA standards) are not reaching national air quality standards for ozone [51]. Los Angeles experienced an average of 103 nonattainment days per year from 2000–2014 to recent increases of 107 nonattainment days in 2015 and 104 nonattainment days in 2017 [51].

California air pollution results from, but is not limited to, higher traffic-related emissions, frequent wildfires, an increase in oil and gas extraction, higher temperature, and topography [51]. Low income populations bear the greatest burden of environmental pollution [51,52]. Low income neighborhoods are more likely to be located near highways and factories. This increased burden of exposure can be measured by the air toxic respiratory hazard index, diesel particulate matter (PM) level in air, PM_2.5_ levels in air, and inhalation of air toxins against percent low-income and percent minority [51]. While urbanization has led to an increased ground-level ozone impacting primarily urban areas, weather conditions and topography can result in impact in rural regions such as California’s primarily rural Central Valley [51].

Many studies have suggested the implications of air pollution on health via epigenetic regulation, but further investigation must be done to examine whether early life exposure to air pollution poses greater risks of breast cancer susceptibility and development. A large component of urban air pollution results from traffic-related air pollution (TRAP). TRAP is a source of particulate matter including black carbon, heavy metals, sulfur dioxides, nitrogen oxides, and polycyclic aromatic hydrocarbons (PAH) and contributes to generating ozone. Heavy metals and PAH are associated with DNA hypomethylation [53]. Ozone can also damage human mammary tissue by contributing to oxidative stress and activating tissue-inflammation [53,54,55].

### 4.3. Cosmetics

The United States Food and Drug Administration (FDA) regulates both drugs and cosmetics. However, FDA regulation of cosmetics is significantly less stringent. Cosmetics are defined by the FDA as “articles intended to be rubbed, poured, sprinkled or sprayed on, introduced into or applied to the human body…for cleansing, beautifying, promoting attractiveness, or altering the appearance”. According to the American Cancer Society, unlike drug regulation, the FDA does not require that cosmetic companies test for product safety. Furthermore, the Federal Food, Drug and Cosmetic Act prohibits the FDA from (1) reviewing ingredients used in cosmetic products prior to marketing and (2) recalling toxic products [51,56,57,58].

From 2004 to 2017, 10,726 adverse event cases have been reported to the FDA [58]; further, 41% of these cases (4427) were cancer-related adverse events associated with cosmetics. Despite the lack of research behind toxicity in cosmetic products, common cosmetics and personal care products have been found to contain chemicals that increases an individual’s risk for contact dermatitis, birth defects in pregnant women, hormone disruption in children and adolescents, and even cancer [51,59,60,61]. These chemicals include but are not limited to diethanolamine (DEA), phthalates, formaldehyde, parabens, bisphenol A (BPA), ethylene oxide and aluminum salts.

The incomplete regulation of cosmetics by the FDA and widespread availability of cosmetics containing potential endocrine disrupters and potential carcinogens is cause for concern. The FDA has made the Center for Food Safety and Applied Nutrition’s Adverse Event Reporting System (CAERS) publicly available in 2016 to increase transparency and encourage adverse event reporting from consumers related to cosmetics [58]. Another cause for concern is that cosmetics can remain in the body long after the initial exposure. The accumulation of toxins during a woman’s lifetime has the potential to impact the fetus during pregnancy and early life use of cosmetics has the potential to also impact young women entering puberty.

#### 4.3.1. Formaldehyde

Hair straightening treatments are known to contain formaldehyde. The amount of formaldehyde in each product varies. Oregon’s Occupational Safety and Health Safety (OSHS) found unsafe levels of formaldehyde in a brand of Brazilian straightener [56]. There are many cases of false advertising, where products labelled “formaldehyde-free” and “salon safe” still contained unsafe levels of formaldehyde.

#### 4.3.2. Aluminum Salts

Aluminum salts are commonly found in antiperspirants. Daily use of antiperspirants results in long-term, local exposure of the breast to aluminum salts. Chronic exposure to aluminum salts has been shown promote DNA damage in animal and human cell culture models. The concern is that chronic exposure to aluminum salts before and during puberty might have the potential to generate genomic instability in rapidly changing breast tissue [60].

#### 4.3.3. Endocrine Disrupting Chemicals (EDC)

Exposure to high levels of estrogenic compounds, or EDCs, impact the interaction of endogenous estrogen and progestogen with their receptors. EDCs plays a role in regulating growth and development by binding to hormone receptors in the cell and blocking or altering downstream cell signaling. Many synthetic compounds are either known or suspected to be EDCs [62]. Known EDCs include bisphenol-A (BPA), parabens, and phthalates [62].

There is increasing evidence that EDCs may reprogram normal progenitor cells in the breast; subsequently, these reprogramed normal cells are transformed by subsequent hormone exposures [61]. Consistent with the high-risk field hypothesis [1], the number of mammary progenitors expands before and during puberty and distribute themselves throughout the ductal tree [60].

BPA is an EDC used to soften plastics and increase its pliability. BPA has been detected in human blood, the placenta, fetal liver, and breast milk. BPA exposure is associated with early puberty and disrupts hormone signaling. In vitro, BPA upregulates AKT to increase proliferation and apoptosis-resistance. In a recent study, 27 genes commonly found to be dysregulated in ER-positive and ER-negative breast cancer interact with EDCs.

Parabens are preservative that are used to extend shelf-life of commercial products. Parabens are found in body creams, antiperspirants, sunscreen, lotion, shampoos [63,64]. There is an association between increased use of paraben-containing body care products in the Western world and the increasing incidence of breast cancer [63,64]. Parabens have been suggested as the agents in body care formulations potentially involved in breast cancer [63,64].

Phthalates are found in hair cosmetics, deodorants, nail polish, and lotions [65,66]. The major pathways of exposure are inhalation, absorption through the skin, and oral intake [66]. Phthalates are known to impair endocrine function via the steroid receptors [67,68]. These interactions are associated with early onset of puberty in women and infertility.

## 5. Nutrition, Obesity, Risk

Many studies have investigated whether diets/nutrients alter breast cancer risk. Preclinical studies provide evidence that specific dietary components may have either a protective or promoting effects [69]. The majority of epidemiologic studies have not definitively shown that any dietary component causes or protects against cancer [69]. Nutritional factors associated with increased breast cancer risk include the consumption of fat, meat, dietary fiber, and alcohol but not all studies confirm this risk. Dietary compounds associated with a decrease risk of breast cancer include vitamin D, folate/B12, carotenoids, and dietary fiber.

The Nurses’ Health Study II (NHSII) investigated the potential association between adolescent dietary patterns and premenopausal breast cancer (614 cases) [70]. A marginal inverse association was observed between the ‘prudent’ dietary pattern (high intake of vegetables, fruits, legumes, fish and poultry) and premenopausal breast cancer; the inverse association was strongest for ER/PR-/- breast cancers [70]. NHSII also showed a positive association between high dietary fiber intake during adolescence and a decrease in breast cancer risk (RR for highest versus lowest quintile 0.84; 95% CI 0.70–1.01; Ptrend = 0.04) [71]; however, women with low fiber intake during adolescence were more likely to smoke, drink alcohol, and have high adulthood BMI [71]. These findings were confirmed by a recent meta-analysis [72]. While these studies are of high quality, they also highlight the difficulty in assigning a mechanistic link between specific dietary components and breast cancer risk; individuals with nutrient-poor diets may also have additional risk factors for cancer such as high alcohol intake, cigarette use, and increased environmental burden.

### 5.1. Obesity, Breast Cancer, Controversy

Obesity was once associated with wealth; today, it is associated with poverty. Disparities in access to healthy food sources (food deserts) [73] and the lack of safe places to exercise all contribute to the obesity epidemic (for a review, see Dietze et al. [74]). Many studies have investigated the relationship between obesity and breast cancer subtypes in individuals of diverse race and ethnicity; while the studies were of high quality, results are highly variable [74]. The variability of results highlights the complex relationship between obesity and breast cancer, especially when comparing individuals of different races and ethnicities.

The relationship between BMI and obesity varies significantly between individuals of different races and ethnic groups [75,76]. For example, self-identified African-Americans/Blacks have a higher muscle mass than non-Hispanic Europeans and Asians [77,78]. As a result, African-Americans are inappropriately categorized as obese. BMI also impacts individuals of different races/ethnicities disproportionately [79,80]. The incidence of type-2 diabetes in non-Hispanic White individuals with a >30 kg/m^2^ was equivalent to the incidence of type-2 diabetes at a BMI of (1) 24 kg/m^2^ in South-Asians, (2) 25 kg/m^2^ in East-Asians, and (3) 26 kg/m^2^ in Black/African-Americans [81]. These studies provide evidence that other factors beyond BMI, such as body composition and/or metabolic health have an impact on breast cancer risk.

### 5.2. Energy-Dense Foods and Risk

The consumption of energy-dense foods is strongly associated with an increased risk of cardiovascular disease, diabetes, and obesity; recent studies provide a potential link with cancer. Energy-dense foods include processed foods containing high fat and/or sugar. In a case-control study with 1692 African American/Black women aged 20 and older (803 cases and 889 controls), frequent energy-dense, fast food consumption was related to increase premenopausal breast cancer risk and with ER+ breast cancers [82].

### 5.3. Diet Soda

The American Heart Association lists soft drinks as among the major sources of dietary sugar [51]. To reduce sugar intake, many health care practitioners recommend non-nutritive sweeteners, including Aspartame, Acesulfame-K, Neotame, Saccharin and Sucralose; these compounds have been labeled by the Food and Drug Association (FDA) as “Generally Recognized as Safe” [51]. The safety of these compounds, however, have been recently called into question. In recent studies, the consumption of artificially sweetened soft drinks was associated with an increased risk of death from cardiovascular disease [83]. Children who consumed artificially sweetened beverages consumed more calories than children who drank sugar-sweetened beverages [84]. There are also concerns that non-nutritive sweeteners may also alter the gut microbiota and increase obesity-associated inflammation and innate and cell-mediated immunity [85].

## 6. Stress

Stress is a multifaceted term that can refer to a stressful stimulus (e.g., the stressor), an organism’s response, either psychological or physiological, and the consequences of this response [86]. The hypothalamic–pituitary–adrenal axis plays a central role in coordinating the physiological stress response. Hypothalamic–pituitary–adrenal axis dysregulation is associated with environmental stressors and predicts behavioral and physical health vulnerability. In animal models, high maternal care is associated with significantly reduced levels of hypothalamic–pituitary–adrenal axis reactivity and lower levels of fearfulness and anxiety. In humans, both hyper and hypo levels of hypothalamic–pituitary–adrenal axis activity are associated with environmental stressors. Hypothalamic–pituitary–adrenal axis reactivity is blunted in women who report childhood physical abuse [87]. Hypercortisolism is evidenced in traumatized and chronically stress individuals [88], and hypercortisolism has been implicated in major depression [89].

Stress researchers have identified “sensitive” and “critical” phases of development; on-going studies provide evidence that early life stressors have a lifelong impact on gene expression [90]. Early life stressors in mice lead to sustained hypothalamic–pituitary–adrenal axis activity and hypomethylation of the hypothalamic arginine vasopressin and pituitary pro-opiomelanocortin gene, as well as hypermethylation of the Nr3c1 control element in turn preventing the up-regulation of the corticotropin-releasing hormone [91].

## 7. Neighborhoods

Neighborhoods are a major determinant of health and life expectancy; individuals living just a few streets apart may have a vastly different risk of environmental exposures, diet, and disease, because of where they live [92,93]. In 2019, significant gaps in health and life expectancy continue to persist across neighborhoods in the United States. For an interactive map of the latest estimates of life expectancy versus census track, please see this recent report [92].

Since neighborhoods matter, it is important to have a detailed understanding of the environments people live and work in. Microscale environmental data are needed to evaluate how neighborhoods may be linked with disease, including street features (sidewalks, safety, street crossings), availability of stores and restaurants (grocery stores, liquor stores, fast food restaurants), crime, sources of pollution (air, water, toxic waste sites), and location of schools and medical facilities.

A better understanding of neighborhood inequities and how they lead to increased breast cancer risk is essential for developing policy that can equitably reduce cancer risk. Environmental justice mapping studies have been particularly useful to encourage the implementation of equitable policies and empower vulnerable communities [94,95,96,97].

Currently, there is a significant movement underway to develop interactive research tools that can support public health advocacy and environmental justice. Examples include (1) the United States Department of Agriculture (USDA) food desert locator [98], (2) Environmental Protection Agency environmental justice tools [51], and (3) some of the dynamic neighborhood/environmental mapping work performed at the University of California, San Diego [99,100,101].

## 8. Conclusions—Call for Advocacy

Early exposure to environmental carcinogens, endocrine disruptors, and unhealthy foods (refined sugar, processed fats, and food additives) are hypothesized to promote molecular damage that increases breast cancer risk; however, as outlined above, evidence for mechanistic causality is lacking. It is difficult to prevent environmental exposures during puberty; young women are repeatedly exposed to media messaging that promotes unhealthy foods.

Young women living in disadvantaged neighborhoods experience additional challenges including a lack of access to healthy food (food deserts) and contaminated water and soil (neighborhood red-lining). Through our K-12 community based science technology, engineering, and math (STEM) programs, we aim to engage elementary/middle-school girls in community science. Students will measure environmental toxins in their tap water, read food labels, and identify food additives; we will work with students to develop strategies to reduce exposures (Figure 4). With our scientific team students, we will (1) test water and soil, (2) teach about diet, and (3) work with the government to develop an action plan. The ultimate goal of our efforts is to foster a generation of young women who are aware of environmental exposures in their environment, have the STEM training to measure and evaluate environmental carcinogens, and are empowered to act as advocates for their own health (Figure 4).

Reducing dietary and environmental exposures during puberty requires a step-wise, coordinated effort. Importantly, this effort must engage multiple stakeholders, neighborhoods, schools, advocacy groups, and the government. We will first test our program on a small scale, leveraging our established Los Angeles STEM programs, epigenetic science, on-going community outreach, water and soil assessment programs, and working relationship with local schools and the government. The effectiveness of our initial study will be evaluated and fine-tuned by a working group consisting of external scientific advisors, breast cancer advocates/community members, local schools, and the government.

## Figures and Tables

**Figure 3 ijerph-17-00493-f003:**
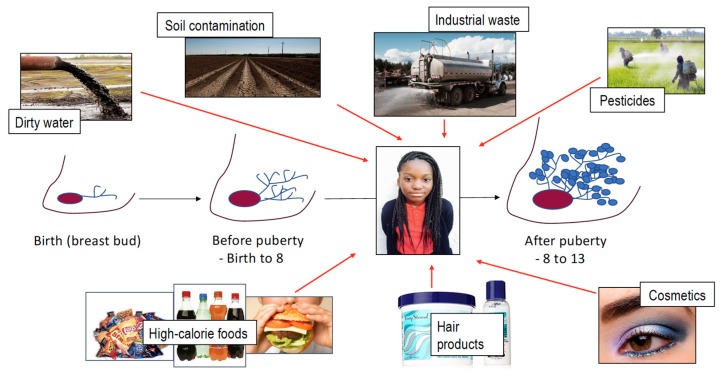
Young women are bombarded with environmental exposures before and at the time of puberty that include polluted water, contaminated soil, industrial waste, pesticides (in water, food, soil), hair care products, and cosmetics.

**Figure 4 ijerph-17-00493-f004:**
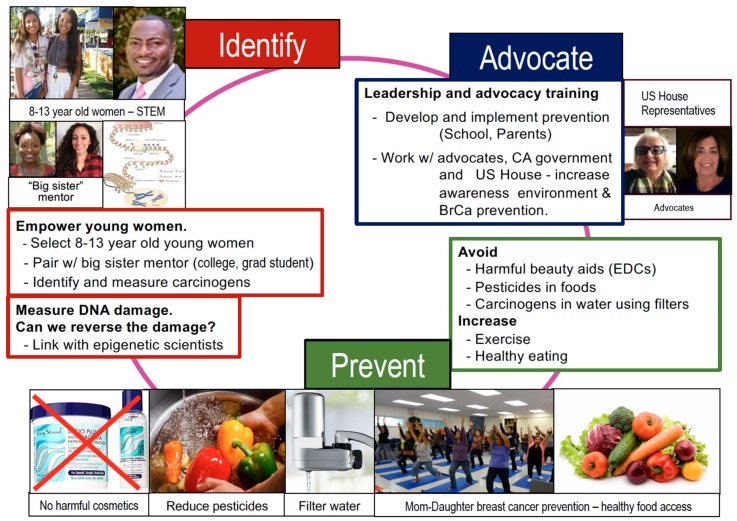
Identify, Prevent, Advocate. Here, our team of scientists, teachers, advocates, and state representatives aim to work together to mentor young women to reduce toxic exposures during puberty and promote healthy diets and exercise within their families and communities. IDENTIFY: We will use our established science technology, engineering, and math (STEM) program to allow young women to work with our scientists to measure chemicals in their environment. PREVENT: We will work with community advocates school and environmental state officials to develop ways to avoid toxic exposures. ADVOCATE: We will work with state and national officials to help our young women present their findings to our state legislature and US House of Representatives. In doing so, our team will help mentor young women who are scientifically trained and will provide our next generation of advocates.

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
