# Peer review of "Environmental Exposures during Puberty: Window of Breast Cancer Risk and Epigenetic Damage"

_ijerph, 2020, doi:10.3390/ijerph17020493_

Round 1

Reviewer 1 Report

This article by Natarajan E et al, review the potentially increased risk of breast cancer in young women, based on the molecular damage that cause the early exposure to environmental carcinogens, endocrine disruptors, and unhealthy foods. It is straightforward, well written, and concise. Definitely deserves to be published and is a valuable contribution to the “International Journal of Environmental Research and Public Health”. Some minor flaws need to be addressed before publication.

Minor points:

[1] Introduction, Line 41: Furthermore, it is interesting from the social point of view, that cancer diagnosed during pregnancy is likely to rise since the delay of childbearing to a later reproductive age. Among maternal malignancies, most common is the breast cancer.

Relevant reference: Boussios S, et al. A review on pregnancy complicated by ovarian epithelial and non-epithelial malignant tumors: Diagnostic and therapeutic perspectives. J Adv Res. 2018 Mar 6;12:1-9.

[2] Epigenetics, Line 25: Please, add here that it has not yet been clarified whether the biological effects of harbouring somatic BRCA1/2 mutations is identical to their germline counterparts (BRCAness phenomenon). In addition, homologous recombination deficiency can result from epigenetic processes leading to silencing of homologous recombination genes, including BRCA1/2.

[3] Epigenetics, Line 51: The paragraph entitled “Epigenetic changes associated with breast cancer” is incorporated twice in the section “Epigenetics”. Please, rephrase accordingly the second one. You can mention that early studies have shown significant efficacy for PARP inhibitors in patients with BRCA1/2 mutations. It has also become evident that BRCA wild-type patients with other defects in the homologous recombination repair pathway benefit from this treatment. With this regards, companion homologous recombination deficiency scores are being developed to guide the selection of patients that are most likely to benefit from PARP inhibition.

Relevant reference: Boussios S, et al. PARP Inhibitors in Ovarian Cancer: The Route to "Ithaca". Diagnostics (Basel). 2019 May 18;9(2). pii: E55.

Author Response

We thank the reviewers for their thoughtful comments. Guided by reviewer comments, the manuscript has been revised

Concern 1: Introduction, Line 41: Furthermore, it is interesting from the social point of view, that cancer diagnosed during pregnancy is likely to rise since the delay of childbearing to a later reproductive age. Among maternal malignancies, most common is the breast cancer.

Relevant reference: Boussios S, et al. A review on pregnancy complicated by ovarian epithelial and non-epithelial malignant tumors: Diagnostic and therapeutic perspectives. J Adv Res. 2018 Mar 6;12:1-9.

Response 1: We agree that this point should be included. In the revised manuscript we include this point and cite the suggested reference [page 2; lines 47-49].

 Concern 2: Epigenetics, Line 25: Please, add here that it has not yet been clarified whether the biological effects of harboring somatic BRCA1/2 mutations is identical to their germline counterparts (BRCAness phenomenon). In addition, homologous recombination deficiency can result from epigenetic processes leading to silencing of homologous recombination genes, including BRCA1/2.

Response 2: We agree with Reviewer 1; the above point is added to the revised manuscript [page 3; lines 5-9].

Concern 3: Epigenetics, Line 51: The paragraph entitled “Epigenetic changes associated with breast cancer” is incorporated twice in the section “Epigenetics”. Please, rephrase accordingly the second one. You can mention that early studies have shown significant efficacy for PARP inhibitors in patients with BRCA1/2 mutations. It has also become evident that BRCA wild-type patients with other defects in the homologous recombination repair pathway benefit from this treatment. With this regards, companion homologous recombination deficiency scores are being developed to guide the selection of patients that are most likely to benefit from PARP inhibition. Relevant reference: Boussios S, et al. PARP Inhibitors in Ovarian Cancer: The Route to "Ithaca". Diagnostics (Basel). 2019 May 18;9(2). pii: E55.

Response 3: We removed the duplicate paragraph. The above section and reference has been added to the revised manuscript [page 3; lines 1-9].

Reviewer 2 Report

This manuscript is an extensive review article that includes a comprehensive set of risk factors and mediating mechanism for breast cancer with special emphasis of environmental exposures during pregnancy. It is an important issue that the paper deals with.

Major concerns:

The strategy for including different factors in the review has not been given. Is this a systematic review, where search strings can be described or it is merely the opinion of the experts? I do not have the necessary time to do bibliographic searches just now, because of being in an international meeting during the last week.  Empowering young woman is included in the title, but the paper do not expand on this issue. It is merely mentioned in the concluding sextion and a figure is included without any references or explanatory text. My suggestions is either change the title of the paper or to treat this issue in a more proper way.

Minor concerns:

Page 2, line 10: what does "a reiew of studies" stand for? Please add information Page 8, line 23: references are lackeing with ragards tgop Parabens. Page 8, line 30: Should be "impair" not "Impair" References 37,38, 54, 80. Is it proper to cite results fron non-peer-review journals in a scientific paper? We do not allow our undergraduate students to use such references? Please amend or add further information. 

Author Response

We thank the reviewers for their thoughtful comments. Guided by reviewer comments, the manuscript has been revised

Concern 1: The strategy for including different factors in the review has not been given. Is this a systematic review, where search strings can be described or it is merely the opinion of the experts?

Response 1: This is the opinions of experts; we gathered these experts to consider ways to reduce exposures in young women before and during puberty. We clarified this issue in the manuscript “Here, we have gathered together a panel of experts discuss some of the potential linkages between environmental exposures, diet, and breast cancer risk.  It is the authors’ hope that this discussion can promote strategic interventions to reduce breast cancer risk in young women.” [page 2; lines 7-10].

Concern 2: Empowering young woman is included in the title, but the paper do not expand on this issue. It is merely mentioned in the concluding section and a figure is included without any references or explanatory text. My suggestions is to either change the title of the paper or to treat this issue in a more proper way.

Response 2: We agree with Reviewer 2; the title has been modified to exclude the words “empowering young women”.

Concern 3: Page 2, line 10: what does "a review of studies" stand for? Please add information.

Response 3: We clarified that that “Here we discuss some of the potential linkages between environmental exposures, diet, and breast cancer risk.”

Concern 4: Page 8, line 23: references are lacking with regards to Parabens.

Response 4: References have been added for both Parabens and Phthalates.

Concern 5: Page 8, line 30: Should be "impair" not "Impair"

Response 5: This error has been corrected.

Concern 6: References 37,38, 54, 80. Is it proper to cite results from non-peer-review journals in a scientific paper? We do not allow our undergraduate students to use such references. Please amend or add further information. 

Response 6:  We agree with Reviewer 2; to this end we amended reference or clarified why specific reports (e.g. United Nations Report) were cited.

Reference 37: Here we quote United Nations Secretary-General Ban Ki-moon. We cite the United Nations report of Ban Ki-moon’s speech. We were not clear in stating our intentions to quote Ban Ki-moon; this is now clarified.

Reference 38: 1) water contamination and lead levels in Flint: a peer review reference is cited, 2) water contamination in Gaza: to our knowledge this issue has not been reported in a peer reviewed article (heavy metal levels have been reported in a peer reviewed manuscript for the remainder of the Palestinian territories but this studies did not include Gaza); here we cite a report from the Rand corporation.

Reference 54: Citation is replaced by a peer reviewed article [Macon et al J. Mammary Gland Biol. Neoplasia, 2013].

Reference 80: A Center for Disease Control report is added; the original citation 80 is an interactive mapping tool developed by the Robert Wood Johnson Foundation; we feel that Robert Wood Johnson reports provide a high degree of accuracy, rigor, and reproducibility.

Round 2

Reviewer 2 Report

Dear authors,

The manuscript is now much better and I agree that the amendment of the titel was a good decision. The empowerment of women needs a more detailed presentation.

Good luck with your further research and development work

Best regards